# Impact of Single- Versus Multiple-Type HPV Infections on Cervical Cytological and Histological Abnormalities: The Dominant Oncogenic Potential of HPV16 Single-Type Infections

**DOI:** 10.3390/diagnostics15222880

**Published:** 2025-11-13

**Authors:** Sunhwa Baek, Sebastian Ludwig, Sophie Lee Sievers, Thomas Einzmann, Yue Zhao, Henryk Pilch

**Affiliations:** 1Department of Obstetrics and Gynecology, University Hospital Cologne and Medical Faculty, 50937 Cologne, Germany; sebastian.ludwig@uk-koeln.de (S.L.); henryk.pilch@uk-koeln.de (H.P.); 2Department of General, Visceral, Cancer and Transplantation Surgery, University Hospital Cologne and Medical Faculty, 50937 Cologne, Germany; yue.zhao@uk-koeln.de

**Keywords:** human papillomavirus (HPV), single-type infection, multiple-type infection, HPV16, alpha-7 clade, alpha-9 clade

## Abstract

**Background:** Persistent infection with high-risk human papillomavirus (HR-HPV) is the primary cause of cervical intraepithelial neoplasia (CIN) and cervical cancer. While HPV testing has become central to screening programs and the frequency of detecting multiple HPV genotypes has subsequently risen, the clinical relevance of multiple-type (MT) HPV infections remains uncertain. This study aimed to investigate the correlation between HPV infection type and the severity of cervical cytological and histological abnormalities. **Methods:** This retrospective study analyzed 340 women with dysplasia and 82 with histologically confirmed cervical cancer treated at the University Hospital Cologne between 2016 and 2019. HPV genotyping was performed using a DNA microarray detecting 41 HPV genotypes. Associations between infection patterns and cytological and histological findings were evaluated. **Results:** Multiple infections accounted for 42% of HPV-positive cases (119 among 284), showing a bimodal age distribution with peaks in patients ≤19 and ≥60 years. HPV16 and HPV18 were most frequently detected in worse than CIN3 lesions (CIN3+), mainly as single-type (ST) infections. Women with ST infections had a significantly higher risk of CIN3+ compared to those with MT infections (*p* = 0.004). HPV16 ST was significantly associated with CIN3+ compared to other HR-HPV ST (*p* = 0.046), whereas MT including HPV16 did not increase CIN3+ risk (*p* = 0.124). Co-infections involving alpha-9 clade types were associated with higher CIN3+ risk, while alpha-7 co-infections did not show an additive effect. Furthermore, coinfections involving two different alpha-9 or -7 were observed infrequently. **Conclusions:** Single HR-HPV infections are more strongly associated with high-grade cervical lesions than multiple infections, especially when HPV 16 is involved. These findings underscore the dominant oncogenic potential of HPV16 and suggest that intergenotypic interactions in MT infections may mitigate malignant progression.

## 1. Introduction

Persistent infection with high-risk human papillomavirus (HR HPV) is the primary cause of cervical intraepithelial neoplasia (CIN) and cervical cancer [1,2]. To date, more than 200 HPV genotypes have been identified, of which approximately 40 infect the genital tract [3,4]. Among these, 12 genotypes are classified by the International Agency for Research on Cancer (IARC) as high-risk (Group 1) due to their established oncogenic potential, followed by probable high-risk HPVs (Group 2A) and possible high-risk HPVs (Group 2B) [5]. HR HPVs belong predominantly to the alpha genus, with phylogenetic subgroups alpha-9 (HPV 16, 31, 33, 35, 52, 58 and 67) and alpha-7 (HPV 18, 39, 45, 59, 68 and 70) being the most frequently associated with cervical carcinogenesis [6].

As HPV testing becomes central to cervical cancer screening, the frequency of detecting multiple-HPV genotypes has risen due to both methodological advancements and population-related factors, including age, immunocompetence, and sexual behavior [7,8,9,10]. Co-infection with more than one HR HPV type is reported in 20–50% of HPV-positive cases and is particularly frequent in younger populations and among those with impaired immune system [11,12,13,14]. Despite this growing prevalence, the clinical significance of multiple type (MT) infections remains unclear [15]. Previous studies have reported inconsistent results regarding the risk caused by MT infections. Several studies demonstrated that coinfections may lead to synergistic interactions, potentially accelerating progression to high-grade lesions and cervical cancer [16,17,18,19]. Moreover, MT infections have been shown to contain more carcinogenic E6/E7 HPV mRNA than a single-type (ST) infection [20]. On the contrary, other studies have reported that MT infections do not exert synergistic or additive effect compared to ST infections [13,21]. Some evidence even suggests that intergenotypic competition or enhanced immune responses caused by MT infections could mitigate the oncogenic potential of individual types, resulting in reduced risk [12,22]. The oncogenic behavior of each HPV types may also vary depending on whether they are present alone or in combination with others, which further complicates interpretation [16,22,23,24,25].

While infections involving phylogenetically related types, particularly within alpha-9 or alpha-7 clades, have been proposed to increase the risk of cervical disease, the potential impact of multiple infections within these clades has not been consistently observed [24,26]. The relationship between HPV infection patterns among clades and their cytological and histological correlations remains insufficiently understood, which poses challenges for clinical decision-making in screening and follow-up procedures.

Considering these uncertainties, we investigated the correlation between HPV infection type focusing ST versus MT infections and the severity of cervical abnormalities assessed by both cytology and histology. Our findings aim to clarify the prognostic relevance of MT infections in cervical disease and support the development of evidence-based recommendations for HPV-based screening strategies.

## 2. Materials and Methods

### 2.1. Study Population

In the present study, 371 women who presented between October 2016 and November 2019 at the dysplasia unit of the University Hospital Cologne were retrospective analyzed. The age of the patients ranged from 18 to 89 years, with a median of 34 years. Of these, 340 patients underwent co-testing with cytology and HPV testing during the same visit as colposcopy and were included in the final analysis. In total, 284 were tested positive for HPV and 56 were HPV-negative. In addition to the dysplasia cohort, a separate group of 82 women with histologically confirmed cervical cancer (CC) from gynecological cancer center between January 2016 and November 2019 was analyzed to investigate HPV type distribution in invasive disease (age range 26–75 years, with a median age of 43 years). Among them, 64 tested positive for HPV, 14 were HPV-negative, and in 4 cases, HPV status was not determinable (Figure 1). The study was conducted in accordance with the Declaration of Helsinki, and approved by the local Ethics Committee of the Medical Faculty of the University of Cologne (No. 20-1582, 28 April 2021).

### 2.2. Specimen Collection and HPV Genotyping

Cytological samples were collected by a dysplasia-certified gynecologist, using a spatula to sample the ectocervix and a cervical brush for the endocervix. Cells were then transferred onto glass slides, fixed in alcohol, and stained using the Papanicolaou methods. Cytological results were classified according to the Munich III nomenclature, with corresponding Bethesda categories for international compatibility [27]. Colposcopic examinations were conducted under standardized conditions using a Zeiss KSK 150 FC colposcope (Carl Zeiss AG, Oberkochen, Germany). In case of major abnormalities or signs of suspected invasion, targeted biopsy was obtained from the most suspicious areas using biopsy forceps (Seidl Biopsy Forceps ER076R; Aesculap AG, Tuttlingen, Germany). Histological examination was performed in 186 HPV-positive and 17 HPV-negative patients. Biopsy were classified according to WHO system [28]. HPV genotyping was performed using a DNA microarray-based method, which detects DNA from 41 clinically relevant HPV genotypes classified as 12 high risk (HPV16, 18, 31, 33, 35, 39, 45, 51, 52, 56, 58, and 59), 12 probably high risk (HPV26, 34, 53, 66, 67, 68a, 68b, 69, 70, 73, 82IS39, and 82MM4), and 17 low risk (HPV6, 11, 40, 42, 43, 44, 54, 55, 57, 61, 62, 72, 81CP8304, 83MM7, 84MM8, 90, and 91) (VisionArray^®^ HPV Chip 1.0 system, ZytoVision, Bremerhaven, Germany). DNA was extracted, amplified, biotinylated, and subsequently hybridized and detected according to the manufacturer’s protocol. Signal detection and analysis were conducted using microarray analysis software designed for hybridization signal evaluation on compatible array chips [29,30].

### 2.3. Statistical Analysis

For statistical analysis, cytological findings were categorized both into three groups: NILM (negative for intraepithelial lesion or malignancy) [Pap I, IIa, IIg], ASC-US [IIp]/LSIL [IIID1], and ASC-H[IIIp]/HSIL[IIID2, IVa-p, IVa-g, IVb-p, IVb-g, V] and, for chi-square test, dichotomized into low-grade (≤LSIL) versus high-grade abnormalities (≥ASC-H/HSIL). Carcinoma in situ, invasive cervical cancers, atypical squamous cells suggestive of high-grade disease as well as rare glandular abnormalities were considered as HSIL. Histological findings were categorized into five categories (benign, CIN1, CIN2, CIN3, and cervical cancer) and additionally dichotomized into less than CIN2 lesions (≤CIN2, CIN2−) versus worse than CIN3 lesions (≥CIN3, CIN3+). Confidence intervals were calculated to estimate the proportion of multiple infections among all HPV-positive cases. The normal approximation method was used when sample sizes were sufficiently large, while the Clopper-Pearson exact method was applied for small sample numbers to ensure accurate interval estimation. To assess the association between HPV infection patterns (single-type [ST] versus multiple-type [MT]) and cytological as well as histological findings, the chi-square test was employed. All statistical tests were performed using a significance level of 5% (α = 0.05). Results with a *p*-value < 0.05 and <0.001 were considered statistically significant and highly significant, respectively. Data analysis was conducted using Microsoft Excel (Version 2024, Microsoft Corporation, Redmond, WA, USA) and IBM SPSS Statistics Version 26.0 (SPSS Inc., Chicago, IL, USA).

## 3. Results

The distribution of single and multiple HPV infections, as well as the proportion of multiple (MT) and high-risk (HR) HPV infections among all HPV positive patients from the dysplasia unit, was analyzed across six age groups (Figure 2 and Table A1). Among the total of 284 HPV-positive cases, there were 165 (58%) with single infections and 119 (42%) with multiple infections. The highest absolute number of multiple infections was observed in the 20–29 age group (n = 46), followed by the 30–39 group (n = 41). The proportion of multiple infections ranged from 35% to 60% across age groups, with peaks observed among patients aged ≤19 years (51.6%, 95% CI: 14.7–94.7%) and ≥60 years (50.0%, 95% CI: 11.8–88.2%). In the 30–39, 40–49, and 50–59 age groups, the proportion of multiple infections was similar, between 35.1% and 37.3%. Among 284 HPV positive cases, 247 cases (87.0%) were positive for high-risk HPV genotypes (HR-HPV). The highest proportions of HR-HPV among all HPV infections were observed in the 30–39 age group (90.1%), followed by 20–29 (86.5%), and 40–49 (84.2%). Slightly lower proportions were found in the ≤19 (80.0%) and 50–59 (82.4%) age groups. The patients aged ≥60 years showed the lowest HR-HPV/HPV ratio (66.7%).

A total of 250 patients were stratified into six age groups to evaluate the distribution of histological findings (Table 1). Among patients aged 19 years or younger (n = 5, 2.0%), two cases (40.0%) were diagnosed as CIN1, two (40.0%) as CIN2, and one (20.0%) as CIN3. No cervical cancer cases were observed in this group. In the 20–29 year age group (n = 56, 22.4%), benign histology was found in seven patients (12.5%), CIN1 in one patient (1.8%), CIN2 in 14 patients (25.0%), CIN3 in 30 patients (53.6%), and cervical cancer in four patients (7.1%). The largest proportion of patients was found in the 30–39 year group (n = 108, 43.2%). Within this group, 11 patients (10.2%) had benign histology, 10 (9.3%) had CIN1, 16 (14.8%) had CIN2 and 46 (42.6%) were diagnosed with CIN3. This group also had the highest absolute number of cervical cancer cases with 25 (23.1%). Among patients aged 40–49 years (n = 51, 20.4%), benign histology was reported in eight cases (15.7%), CIN1 in five cases (9.8%), CIN2 in four cases (7.8%), CIN3 in 17 cases (33.3%), and cervical cancer in 17 cases (33.3%), showing a significant increase in invasive disease compared to younger age groups. In the 50–59 year age group (n = 21, 8.4%), three patients (14.3%) had benign histology, one (4.8%) had CIN1, one (4.8%) had CIN2, five (23.8%) had CIN3, and 11 (52.4%) had cervical cancer. Among patients ≥60 years (n = 9, 3.6%), one (11.1%) had CIN2, while the remaining eight patients (88.9%) were diagnosed with cervical cancer.

As next, we examined how cytological findings correspond with histological diagnoses among patients from the dysplasia unit (Table 2). Cytological findings were categorized into three groups: NILM, ASC-US/LSIL, and ASC-H/HSIL. Of the twelve patients (6.5%) with benign cytology, six patients (50.0%) had benign histology, two patients (16.7%) had CIN1, one patient (8.3%) had CIN2, and notably, three patients (25.0%) presented CIN3. No cases of cervical cancer were identified in this group. In the ASC-US/LSIL group (n = 20, 10.8%), histological evaluation revealed benign findings in seven patients (35.0%), CIN1 in two patients (10.0%), CIN2 in seven patients (35.0%), and CIN3 in four patients (20.0%). Again, no cervical cancer cases were observed in this category. The majority of patients (n = 154, 82.8%) were classified cytologically as ASC-H and HSIL. Among these, histology showed benign results in 16 cases (10.4%), CIN1 in 15 (9.7%), CIN2 in 27 (17.5%), CIN3 in 94 (61.0%), and cervical cancer in two patients (1.3%).

The genotype-specific distribution of HPV revealed distinct patterns between CIN3+ and CIN2− lesions (Figure 3 and Table A2). High-risk types HPV16 and HPV18 were most frequently detected in CIN3+ cases, predominantly as single infections. HPV16 was the most prevalent genotype across both infection types and lesion severity. In CIN3+ cases, it was detected in 93 cases (39.4%), with 74 (31.4%) as single infections and 19 (8.1%) in combination with other HPV types. For CIN2− lesions, HPV16 was observed in 22 cases (16.4%), of which 15 (11.2%) were single and seven (5.2%) were multiple infections. As second most frequent genotype, HPV18 was identified in 17 infections (7.2%), comprising 13 single (5.5%) and four multiple (1.7%) infections in CIN3+ lesions. For CIN2− lesions, they were found in five cases (3.7%), including three single (2.2%) and two multiple (1.5%) infections.

HPV31 and HPV45 were also more frequently detected in CIN3+ lesions, mainly as multiple infections (n = 14 [5.9%], MT: 12 [5.1%]; n = 11 [4.7%], MT: 7 [3.0%], respectively). In contrast, among CIN2− lesions, four cases (3.0%) were attributed to HPV31, with three as single infections (2.2%) and one as a multiple infection (0.8%), whereas HPV45 was observed only as multiple infections (n = 3 [2.2%]). HPV52 showed a similar distribution in CIN3+ (n = 14 [5.9%], MT: 12 [5.1%] vs. ST: 2 [0.8%]) and CIN2− lesions (n = 7 [5.3%], MT: 6 [4.5%] vs. ST: 1 [0.8%]), with a predominance of multiple infections in both groups.

In contrast, other high-risk and probably high-risk genotypes were more frequently found in CIN2− lesions, predominantly as multiple infections.

The correlation between various HPV infection types and severity of cytology and histology was examined (Table 3). There was no statistically significant association between HPV infection type and cytology. For instance, the distribution of cytological findings (NILM/ASC-US/LSIL vs. ASC-H/HSIL) did not significantly differ between ST and MT infections (*p* = 0.815), nor between HR HPV ST and MT (*p* = 0.296). Similarly, no significant associations were observed for HPV16 ST versus MT (*p* = 0.582), HPV16 ST versus HR HPV ST without HPV16 (*p* = 0.232), or MT including HPV16 compared to other HR HPV MT (*p* = 0.378).

In contrast, histological abnormalities showed some significant associations with specific HPV infection patterns. Women with single-type HPV infections had a significantly higher risk of CIN3+ lesions compared to those with multiple infections (χ^2^ = 8.1311, *p* = 0.004). Similarly, HR HPV ST were significantly associated with more severe histological abnormalities compared to MT (χ^2^ = 7.118, *p* = 0.008). While HPV16 ST also showed a significant association with CIN3+ histology compared to other HR HPV ST (χ^2^ = 3.975, *p* = 0.046), MT including HPV16 displayed a non-significant trend toward CIN3+ lesions compared to HR HPV MT without HPV16 (*p* = 0.124). The risk of CIN3+ lesions from HPV16 ST was not significantly higher than from MT including HPV16 (*p* = 0.251). HR HPV ST versus MT without HPV16 did not reach statistical significance (*p* = 0.163).

The analysis of HPV clades-specific combinations revealed that multiple HPV infections with alpha-9 clade are associated with an increased risk of CIN3+. The frequency of CIN3+ cases was similar when alpha-9 clade was co-infected with either alpha-7 clade (n = 15) or non-high-risk types (non-a7/9, n = 15), suggesting that alpha-9 HPV clades act as the primary risk factor. The presence of alpha-7 alongside alpha-9 does not appear to confer additional risk for CIN3+ beyond that of alpha-9 alone. Notably, there were no combinations between alpha-7 clades. still more common in CIN3+ cases (n = 8) than in Co-infections involving two different alpha-9 types were observed infrequently, CIN2− cases (n = 5). In contrast, combinations involving only non-a7/9 types were less frequent in CIN3+ (n = 4) compared to CIN2− (n = 6), indicating a possibly lower oncogenic potential (Figure 4 and Table A3).

## 4. Discussion

The distribution of single-, multiple-, and high-risk HPV infections across age groups shows that ST is more frequent than MT (n = 165 [58%] vs. n = 119 [42%]) and presents an age-dependent bimodal pattern, with peaks among patients aged ≤19 years (51.6%; 95% CI: 14.7–94.7%) and ≥60 years (50.0%; 95% CI: 11.8–88.2%). Although the data on proportion of MT should be interpreted with caution due to the small sample size and wide confidence intervals in patients aged ≤19 and ≥60 years, our results are consistent with those of other studies [22,31,32]. It is well known that younger women show higher rates of multiple HPV infections, which may be attributed to an immature immune system, early onset of sexual activity, and limited prior immunity to HPV [9,33,34]. In older women, the increased incidence of multiple HPV infections is likely attributable to immunosenescence, reactivation of latent infections acquired earlier in life, and cumulative lifetime sexual exposure, with most incident detections reflecting past rather than recent sexual activity [35,36,37]. Furthermore, postmenopausal hormonal changes may further impair local immunity and promote viral persistence [38]. HR-HPV was detected in most HPV-positive cases across all age groups, accounting for 87.0% overall. Notably, an inverse pattern was observed between the proportion of multiple infections and high-risk HPV among all HPV-positive cases indicating that multiple-type infections might be more likely to possess transient or less oncogenic potential, whereas high-risk HPV types are more frequently associated with persistent infections [1].

The distribution of histological diagnoses across different age groups showed age-related trends in cervical lesion severity. Precancerous lesions were more frequently observed in younger women than cervical cancer, suggesting that the onset of cervical abnormalities can occur earlier in life. While no cancer was observed in patients ≤19 years, cancer rates rose from 7.1% (20–29 years) to 23.1% (30–39 years), 33.3% (40–49 years), 52.4% (50–59 years), and 88.9% in patients ≥60 years. The 30–39 age group had the highest absolute number of cancer cases (n = 25), while the highest proportion was seen in the ≥60 group. Our results align with previous studies, which reported that the incidence of CIN2/CIN3 significantly declined with increasing age, while the rate of cervical cancer increased [39]. One possible explanation is the lower participation rate in cervical cancer screening among older women compared to younger ones, reducing the likelihood of timely detection and treatment of precancerous lesions. Additionally, age-related differences in HPV clearance play a crucial role. Older women appear to clear HPV infections less efficiently, potentially due to immunosenescence, which may allow for greater viral persistence and subsequent disease progression [40,41]. This age-related result of HPV and histology underscores the importance of sustained screening across all age groups, including postmenopausal women.

The correlation between cytology and histology demonstrates a strong association between higher-grade cytological abnormalities and more severe histological diagnoses. In total, 61.0% of patients with ASC-H/HSIL cytology were found to have CIN3, and cervical cancer cases occurred only in this group. Nevertheless, the detection of CIN3 in three patients (25%) with benign cytology despite of small number highlights the potential for underdiagnosis based on cytology alone. Notably, NILM/CIN3 cases in our cohort were associated with HR HPV MT (HPV 39/90; HPV 16/83; HPV 16/73/84). Although co-testing and biopsy were performed simultaneously in the present study, the consistent presence of HR HPV in these cases indicates that HR HPV positivity, even in the context of benign cytology, warrants careful evaluation, underscoring the importance of HPV co-testing [42,43].

The HPV genotype-specific analysis of severity of cervical lesions showed that HPV16 and HPV18 were most prevalent in CIN3+ lesions, mainly as single infections. HPV16 was the dominant type across all categories, accounting alone for nearly 40% of CIN3+ cases. HPV31, HPV45, and HPV52 were also more frequent in CIN3+, mostly as multiple infections, while other high-risk and probable high-risk types were more often found in CIN2− lesions, typically as multiple infections as well. Additionally, single infections were significantly associated with an increased risk of CIN3+ compared to multiple infections (*p* = 0.004 for all HPV types; *p* = 0.008 for high-risk HPV). Notably, HPV16 single infections showed a significantly stronger association with CIN3+ lesions than other high-risk single-type infections (*p* = 0.046). In contrast, no significant correlation was observed between HPV infection type and cytological findings. Our findings support a higher oncogenic potential of single infections compared to multiple infections, especially when HPV16 is involved. This aligns with previous studies showing that HPV16 alone confers a greater risk for high-grade lesions than when present in coinfection [21,22,44]. Wentzensen et al. observed that the risk of HSIL was higher in women with HPV16 single infection than in those coinfected with other high-risk types, and that multiple infections involving HPV16 were associated with worse cytological findings compared to multiple infections without HPV16, underscoring its etiological dominance [21]. When HPV16 is present with another carcinogenic HPV, HPV16 is usually the cause in cervical lesions, as the risk of HSIL was unchanged when HPV16 was present alone or in coinfection [44]. Several studies have demonstrated that HPV16 predominantly presents as a ST infection in patients with CIN2+, whereas other high-risk genotypes are more frequently involved in MT infections. Zhong et al. reported that genotypes such as HPV73, 53, and 66 were more frequently detected in MT infections, while HPV16, 18, and 58 were more commonly found as single infections in CIN2+ cases [22]. Similarly, Dickson et al. found that HPV16, 58, and 66 were less likely to occur in MT infections, while types such as HPV52, 53, 81, and 83 were more often involved in MT, indicating differing infection patterns [45]. Consistent with these findings, we observed that patients with a single infection of HPV16 had a significantly higher incidence of CIN3+ compared to those with other HR HPV single infections, while MT including HPV16 did not significantly increase the risk for CIN3+ compared to MT infections without HPV16. This supports that coinfection with other genotypes may mitigate the pathogenic potential of HPV16, possibly due to intergenotypic competition or enhanced immune responses caused by MT [22,24,40]. This interpretation aligns with previous studies reporting no additive or synergistic effect of multiple HPV infections on the development of high-grade cervical lesions [12,21].

The analysis of HPV clades-specific combinations revealed that multiple HPV infections with alpha-9 clades are associated with an increased risk of CIN3+, suggesting the predominant role of alpha-9 types in driving progression to high-grade cervical lesions. The presence of alpha-7 alongside alpha-9 does not appear to confer additional risk for CIN3+ beyond that of alpha-9 alone. As expected, combinations involving only non-a7/9 types were less frequent in CIN3+ compared to CIN2−, indicating a possibly lower oncogenic potential. Coinfections involving two different alpha-9 clades were observed infrequently. Moreover, there were no combinations involving two alpha-7 types. Patterns of HPV coinfection among same clades remain controversial. Some studies reported clustering of phylogenetically related types with genetic similarity in the L1 region (e.g., HPV33/58, 18/45) [46] and suggested that women harboring multiple genotypes with related clades may have a higher risk of cervical cancer than those infected with unrelated clades [24]. However, most evidence indicates that multiple infections occur largely at random, with no consistent preference for alpha-7 or alpha-9 combinations [26,47,48]. The relatively low occurrence of coinfections involving two different alpha-9 or -7 types may reflect biological competition among closely related HPV types, cross-immunity elicited by one alpha-clade [49], or methodological challenges in distinguishing closely related types [46]. Although partial cross-protection has been observed for specific pairs such as HPV31 with 16 and HPV45 with 18 in vaccinated women [13] and reduced viral loads of HPV16/18 in coinfections with related types have also been reported [50], overall there is no clear evidence that coinfection patterns are determined by phylogenetic relatedness. The limited sample size of our study further precludes definitive conclusions.

A limitation of the present study is its small sample size, particularly in extreme age groups, which further restricts the subgroup analyses of interest. Another limitation is the single-center design at a university hospital, where patients were referred following a suspicious examination in a primary care setting, limiting the generalizability of our findings. Despite these limitations, the study provides important insights into young patients, a population for which routine screening is not generally recommended according to most international guidelines. The comprehensive analysis of the correlation between HPV infection type and cervical abnormalities strengthens the clinical relevance of our findings. These observations may contribute valuable information to clarify the prognostic significance of HPV infection types in cervical disease and support the development of evidence-based recommendations for HPV-based screening strategies.

## 5. Conclusions

Understanding the distribution pattern of HPV genotypes is critical for its prevention and vaccine strategy. In our analysis, HPV16 was the dominant type across all categories, showing a significantly stronger association with CIN3+ lesions than other high-risk single-type infections (*p* = 0.046). Our findings support the higher oncogenic potential of single infections compared to multiple infections, especially when HPV16 is involved. Our results show that coinfection with other genotypes may mitigate the pathogenic potential of HPV16, possibly due to intergenotypic competition or enhanced immune responses caused by MT.

## Figures and Tables

**Figure 1 diagnostics-15-02880-f001:**
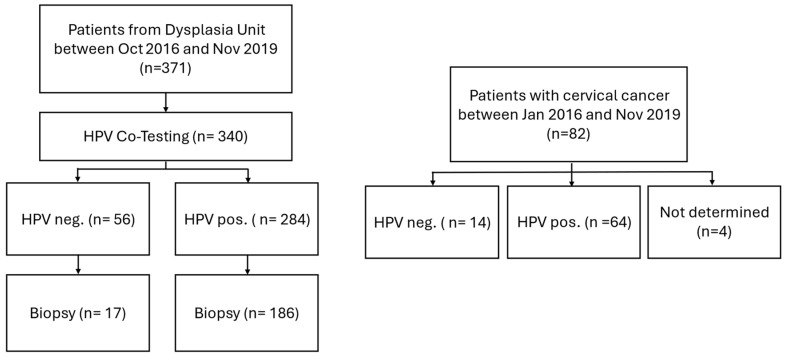
Flow chart of study population.

**Figure 2 diagnostics-15-02880-f002:**
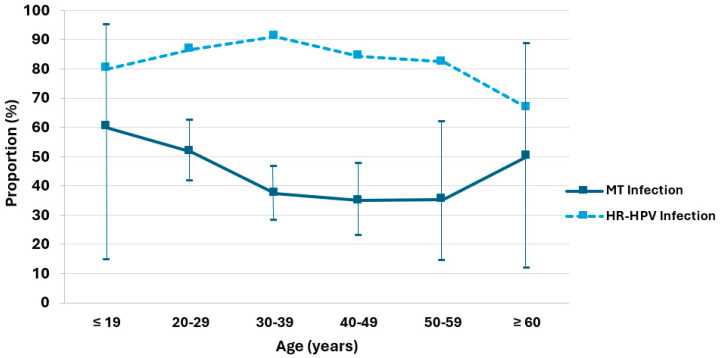
Age-stratified proportions of multiple HPV infections (solid line) with 95% confidence interval and high-risk HPV infections (dotted line) among HPV-positive patients.

**Figure 3 diagnostics-15-02880-f003:**
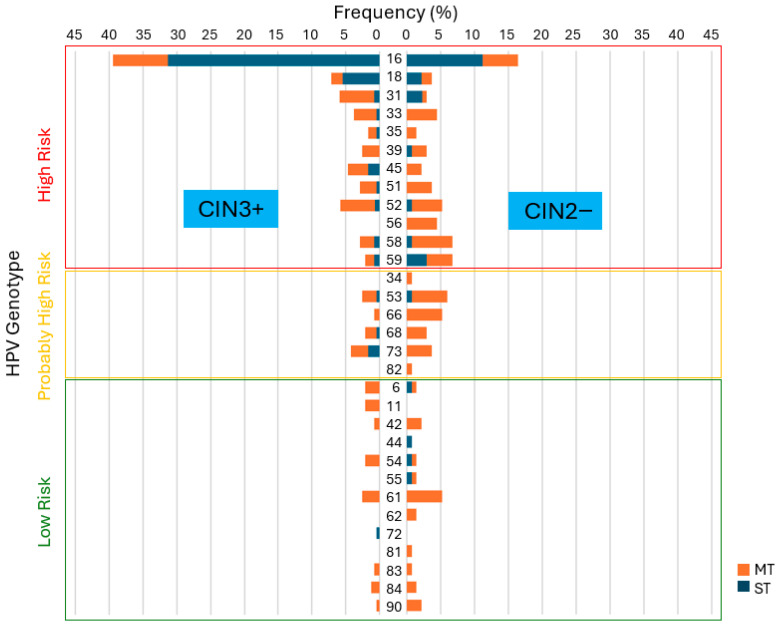
HPV genotype-specific distribution by infection type (MT vs. ST) and severity of lesions (CIN3+ versus CIN2−).

**Figure 4 diagnostics-15-02880-f004:**
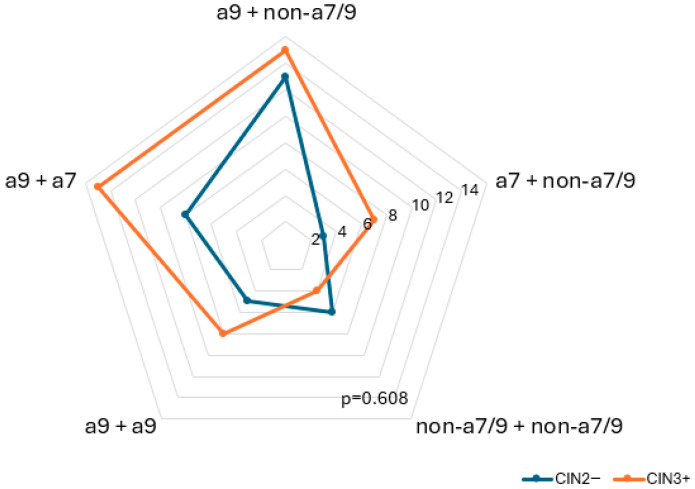
Multiple HPV genotype combinations including alpha-9 (a9), alpha-7 clade (a7), and other types (non-a7/9) in CIN2− and CIN3+ lesions.

**Table 1 diagnostics-15-02880-t001:** Distribution of histology by age group.

	Histology					
Age (Years)	Total (%)	Benign (%)	CIN1 (%)	CIN2 (%)	CIN3 (%)	CC (%)
≤19	5 (2)		2 (40)		2 (40)	1 (20)
20–29	56 (22.4)	7 (12.5)	1 (1.8)	14 (25)	30 (53.6)	4 (7.1)
30–39	108 (43.2)	11 (10.2)	10 (9.3)	16 (14.8)	46 (42.6)	25 (23.1)
40–49	51 (20.4)	8 (15.7)	5 (9.8)	4 (7.8)	17 (33.3)	17 (33.3)
50–59	21 (8.4)	3 (14.3)	1 (4.8)	1 (4.8)	5 (23.8)	11 (52.4)
≥60	9 (3.6)				1 (11.1)	8 (88.9)
Total	250 (100)	29 (11.6)	19 (7.6)	35 (14)	101 (40.4)	66 (26.4)

**Table 2 diagnostics-15-02880-t002:** Correlation between cytology and histological findings.

	Histology					
Cytology	Total (%)	Benign (%)	CIN1 (%)	CIN2 (%)	CIN3 (%)	CC (%)
NILM	12 (6.5)	6 (50)	2 (16.7)	1 (8.3)	3 (25)	
ASC-US/LSIL	20 (10.8)	7 (35)	2 (10)	7 (35)	4 (20)	
ASC-H/HSIL	154 (82.8)	16 (10.4)	15 (9.7)	27 (17.5)	94 (61.0)	2 (1.3)

**Table 3 diagnostics-15-02880-t003:** Correlation between cytology/histology and HPV infection.

	Cytology				Histology			
			χ^2^	*p*			χ^2^	*p*
	N/ACS-US/LSIL	ACS-H/HSIL			CIN2−	CIN3+		
ST	38	126	0.0549	0.815	33	107	8.1311	0.004
MT	29	90			35	49		
HR HPV ST	27	128	1.0919	0.296	28	101	7.118	0.008
HR HPV MT	24	82			29	45		
HPV16 ST	12	73	0.3024	0.582	15	74	1.3187	0.251
HPV16 MT	7	32			7	19		
HPV16 ST	12	73	1.4263	0.232	15	74	3.9753	0.046
HR HPV (HPV16-) ST	15	55			13	27		
HPV16 MT	7	32	0.7758	0.378	7	19	2.3687	0.124
HR HPV (HPV16-) MT	17	50			20	24		
HR HPV (HPV16-) ST	15	55	0.2975	0.585	13	27	1.9474	0.163
HRHPV (HPV16-) MT	17	50			20	24		

N: NILM; ST: single infection; MT: multiple infection; HR HPV: high-risk HPV; HR HPV (HPV16-): HR HPV other than HPV16.

## Data Availability

The data presented in this study are available on request from the corresponding author.

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
