# Peer review of "Impact of Single- Versus Multiple-Type HPV Infections on Cervical Cytological and Histological Abnormalities: The Dominant Oncogenic Potential of HPV16 Single-Type Infections"

_diagnostics, 2025, doi:10.3390/diagnostics15222880_

Round 1

Reviewer 1 Report

Comments and Suggestions for Authors

This retrospective study evaluates the relationship between single-type and multiple-type HPV infections and the severity of cervical cytological and histological abnormalities in a German hospital cohort. The authors conclude that single high-risk HPV infections are more strongly associated with CIN3+ lesions than multiple infections, supporting the dominant oncogenic potential of HPV16.

The topic is relevant to ongoing refinement of HPV-based screening and vaccination strategies. The paper is well-organized, methodologically sound, and well-written, with results that are consistent with international data.

I have a few suggestions for improvement:

The finding that some CIN3 lesions occurred in cytologically benign samples is important. The authors should emphasize its clinical implication for screening (supporting HPV co-testing) and briefly discuss possible reasons (sampling errors, lesion localization, transient regression).

The limitations of the study are mentioned briefly, but I suggest expanding them to include that the study is single-centre, and that the sample sizes are uneven (especially in extreme age groups).

Figure 1. some words are in German (Biopsie, co-testung).

Some minor English language corrections:

  • Line 27 is unclear („most prevalent in severe than CIN3“)
  • ST is not defined but is used in the abstract
  • Line 111 (biopsy instead of biopsies)
  • Line 182 („Patients equal and older than 60 years“ – a symbol can be used for this)

I suggest the paper be accepted after addressing these comments.

Author Response

Comment 1:

The finding that some CIN3 lesions occurred in cytologically benign samples is important. The authors should emphasize its clinical implication for screening (supporting HPV co-testing) and briefly discuss possible reasons (sampling errors, lesion localization, transient regression).

Response 1:

We thank the reviewer´s clinically insightful comment. Indeed, NILM/CIN3 cases in our cohort were associated with high-risk HPV infections, with multiple HPV genotypes present in each patient (HPV 39/90; HPV 16/83; HPV 16/73/84). As only the patients with simultaneous co-testing and biopsy were included in our study, high-risk HPV positivity did not lead to the diagnostic re-evaluation. Nevertheless, our results indicate that the presence of high-risk HPV in patients with benign cytology can guide clinicians to carefully assess for underlying high-grade lesions, supporting the clinical value of HPV co-testing. Regarding the possible reasons for discrepancy, while biopsy is performed generally highly specifically and may not capture the lesion of dysplasia, cytology typically samples a broader cervical area using both spatula and brush. Of course, the possibility of transfer-related artifacts cannot be entirely excluded. Since co-testing and biopsy were performed simultaneously in our study, we cannot provide a comment on transient regression of cytological/histological abnormalities. All authors agree that the importance of HPV co-testing can never be emphasized enough. Despite that, practical considerations such as cost-effectiveness, the high prevalence of transient HPV infections, and resource availability must also be taken into account when designing screening strategies. Since our study included only three cases with discrepancies between cytology and histology and the discrepancy itself is not the primary focus of our study, we chose to limit the discussion of potential sources of error. We hope this approach is in line with the reviewer’s intent. At the same time, following the reviewer’s recommendation, we were able to highlight the importance of HPV co-testing based on our own data, and we are grateful for this valuable suggestion

Comment 2:

The limitations of the study are mentioned briefly, but I suggest expanding them to include that the study is single-centre, and that the sample sizes are uneven (especially in extreme age groups).

Response 2: 

We thank the reviewer for this valuable suggestion. In response, we have added the limitation section to acknowledge the small sample size, particularly in extreme age groups, which limits subgroup analyses, and the single-center design at a university hospital, where patients were typically referred after suspicious findings in primary care, which may limit generalizability.

Comment 3: 

Figure 1. some words are in German (Biopsie, co-testung).

Some minor English language corrections:

  • Line 27 is unclear („most prevalent in severe than CIN3“)
  • ST is not defined but is used in the abstract
  • Line 111 (biopsy instead of biopsies)
  • Line 182 („Patients equal and older than 60 years“ – a symbol can be used for this)

Response 3: 

We appreciate the reviewer’s careful reading and helpful comments. Based on your feedback we corrected the sentences/terms.

Reviewer 2 Report

Comments and Suggestions for Authors

HPV infection is highly prevalent and a necessary cause for the development of cervical cancer, making its detection essential in population screening to identify high-grade lesions early and prevent their progression to cancer. We know that years pass between infection and the development of a lesion, and that cancer appears at least a decade after infection.

This study evaluates multiple infections versus those caused by a single viral type. As reflected in the literature, the most frequently found viruses are types 16 and 18, in both single and multiple infections.

As reflected in the current literature, there is a higher prevalence of HPV infection in young people and, currently, in those over 60 years of age. Some studies also demonstrate an epidemiological peak in the 40s, which is not reflected in this study.

The fact that the study was conducted in young patients is noteworthy. According to most international guidelines, screening is not indicated for women under 25-30 years of age, precisely because of the high viral prevalence and the 90% viral clearance rate. Even the indication for treating cervical lesions before these ages would be debatable. Perhaps this higher prevalence in this group could influence the results obtained regarding the lower incidence of lesions in multiple infections. Could this happen here?

Overall, I believe this study represents a further step in understanding the burden of HPV disease and its evolution. The contribution regarding the lower severity of multiple infections, even when type 16 is present, is of great value. This aspect is relevant and I believe it should be evaluated in future studies.

Zhou D, Xue J, Sun Y, Zhu L, Zhao M, Cui M, Zhang M, Jia J, Luo L. Patterns of single and multiple HPV infections in female: A systematic review and meta-analysis. Heliyo 2024;10:e35736

 Yu Y, Hao J, Mohamed SB, Fu S, Zhao F, Qiao Y. The prevalence of multiple or single HPV infection and genotype distribution in healthy Chinese women: A systemic review. J Can Res Ther 2024;20:1265-73.

Fontham ET, Wolf AMD, Church TR, Etzioni R, et al. Cervical cancer screening for individuals at average risk: 2020 guideline update from the American Cancer Society. Ca Cancer JClin 2020;70:321-346.

Author Response

Comment 1:

The fact that the study was conducted in young patients is noteworthy. According to most international guidelines, screening is not indicated for women under 25-30 years of age, precisely because of the high viral prevalence and the 90% viral clearance rate. Even the indication for treating cervical lesions before these ages would be debatable. Perhaps this higher prevalence in this group could influence the results obtained regarding the lower incidence of lesions in multiple infections. Could this happen here?

Response 1: 

We thank the reviewer for this thoughtful comment. We agree that HPV prevalence and spontaneous viral clearance are particularly high in younger women, which explains why most international guidelines do not recommend routine screening or treatment in this age group. However, we would like to clarify that our study population does not represent a general screening cohort. All patients were referred to our dysplasia unit following suspicious findings, which explains the presence of younger patients in our cohort. Furthermore, in our data, the proportion of multiple infections in the ≤19 and 20–29 age groups was relatively high (51.6% and 60.0%, respectively), and the proportion of CIN3+ lesions in these age groups was also substantial (60.0% and 60.8%). Therefore, we believe that the pathogenic relevance of multiple infections was not underestimated in younger women in our cohort, and the association between multiple HPV types and high-grade lesions was not diminished by age-related transient infection patterns. Nevertheless, due to the small sample size in the younger age group, interpretation should remain cautious. Based on your comment, we have added the limitations section and clarified that the study is constrained by the small sample size, particularly in extreme age groups, which limits detailed subgroup analyses, as well as by its single-center, referral-based design, which may restrict the generalizability of our findings.

Comment 2: 

Overall, I believe this study represents a further step in understanding the burden of HPV disease and its evolution. The contribution regarding the lower severity of multiple infections, even when type 16 is present, is of great value. This aspect is relevant and I believe it should be evaluated in future studies.

Response 2:

We sincerely thank the reviewer for their positive and encouraging comments. We greatly appreciate the recognition of the relevance of our findings. We also thank the reviewer for providing additional references; while these studies did not provide new data directly applicable to our current analysis, we appreciate the guidance and context they offer. We are currently continuing data collection to expand the cohort, and we hope that future analyses with a larger sample size will allow these observations to be further validated and present.